# The Molecular Epidemiology of Clade 2.3.4.4B H5N1 High Pathogenicity Avian Influenza in Southern Africa, 2021–2022

**DOI:** 10.3390/v15061383

**Published:** 2023-06-16

**Authors:** Celia Abolnik, Thandeka Phiri, Belinda Peyrot, Renee de Beer, Albert Snyman, David Roberts, Katrin Ludynia, Frances Jordaan, Michele Maartens, Zehaad Ismail, Christine Strydom, Gerbrand van der Zel, Jade Anthony, Nadine Daniell, Liesl De Boni, John Grewar, Adriaan Olivier, Laura Roberts

**Affiliations:** 1Department of Production Animal Studies, Faculty of Veterinary Science, University of Pretoria, Pretoria 0110, South Africalaura.roberts@westerncape.gov.za (L.R.); 2Provincial Veterinary Laboratory, Western Cape Department of Agriculture, Stellenbosch 7600, South Africa; 3Southern African Foundation for the Conservation of Coastal Birds (SANCCOB), Cape Town 7441, South Africa; 4Department of Biodiversity and Conservation Biology, University of the Western Cape, Bellville 7535, South Africa; 5Assurecloud (Pty) Ltd., Midrand 1683, South Africa; 6SMT Veterinary Laboratory, Irene, Pretoria 0178, South Africa; 7Gauteng Department of Agriculture and Rural Development, Johannesburg 2000, South Africa; 8jDATA (Pty) Ltd., Sandbaai 7200, South Africa; 9South African Ostrich Business Chamber, Oudtshoorn 6620, South Africa; 10Department of Agriculture, Western Cape Government, Elsenburg 7607, South Africa

**Keywords:** high pathogenicity avian influenza, clade 2.3.4.4 H5N1, evolution, phylogenetic analysis, southern Africa, coastal seabirds

## Abstract

In southern Africa, clade 2.3.4.4B H5N1 high pathogenicity avian influenza (HPAI) was first detected in South African (SA) poultry in April 2021, followed by outbreaks in poultry or wild birds in Lesotho and Botswana. In this study, the complete or partial genomes of 117 viruses from the SA outbreaks in 2021–2022 were analyzed to decipher the sub-regional spread of the disease. Our analysis showed that seven H5N1 sub-genotypes were associated with the initial outbreaks, but by late 2022 only two sub-genotypes still circulated. Furthermore, SA poultry was not the source of Lesotho’s outbreaks, and the latter was most likely an introduction from wild birds. Similarly, SA and Botswana’s outbreaks in 2021 were unrelated, but viruses of Botswana’s unique sub-genotype were introduced into SA later in 2022 causing an outbreak in ostriches. At least 83% of SA’s commercial poultry cases in 2021–2022 were point introductions from wild birds. Like H5N8 HPAI in 2017–2018, a coastal seabird-restricted sub-lineage of H5N1 viruses emerged in the Western Cape province in 2021 and spread to Namibia, causing mortalities in Cape Cormorants. In SA ~24,000 of this endangered species died, and the loss of >300 endangered African penguins further threatens biodiversity.

## 1. Introduction

The high pathogenicity avian influenza (HPAI; family *Orthomyxovirdae*) Goose/Guangdong (Gs/GD) lineage of subtype H5Nx viruses emerged in Asian poultry in 1996. By around 2005, the virus had established a reservoir in wild aquatic bird populations and began, in an unprecedented manner, to spread around the globe. It spread through Asia into Europe, the Middle East, Asia-Pacific and the Americas in multiple waves, with the seasonal movements of migratory species [1]. Clade 2.3.4.4 of the Gs/GD H5Nx lineage has been particularly pervasive since 2014, causing catastrophic disease outbreaks in poultry and wild birds, and these outbreaks seem to be increasing in frequency and magnitude [1,2,3]. 

In 2016–2017, a clade 2.3.4.4B H5N8 strain spread extensively through the Middle East and western Eurasia, then to Egypt, Tunisia, Nigeria, Niger, Cameroon and Uganda, before eventually reaching Zimbabwe and South Africa in May and June 2017, respectively [4]. By November 2017 (late spring), the South African epizootic in commercial and backyard poultry, captive and terrestrial wild birds had subsided, but from December 2017 H5N8 HPAI-associated mass mortalities of thousands of Swift Terns (*Thalasseus bergii*) and other coastal seabird species, like the endangered African penguin (*Spheniscus demersus*), began along the southern Cape coastline which lasted until May 2018 [5,6]. A clade 2.3.4.4B H5N8 HPAI outbreak in the African penguin colony on Halifax Island, Namibia, which started months later in December 2018 [7] was phylogenetically linked to the distinctive South African coastal seabird sub-lineage [5]. 

The origin of the current wave of clade 2.3.4.4B H5Nx HPAI, with H5N1 as the dominant subtype, was phylogenetically traced back to northern Europe in October 2020. It has the widest geographic reach to date with outbreaks reported in over 72 countries [3,8]. The first outbreak of clade 2.3.4.4 H5N1 HPAI in Africa was detected in Pout, Senegal in late December 2020 in commercial layer hens, followed in mid-January 2021 by an H5N1 HPAI-associated mass mortality event in Great White Pelicans (*Pelicanus onocrolatus*) in the Djoudji National Bird Sanctuary. The Senegalese outbreaks were phylogenetically linked to those that started in Europe the previous October [9] and reports of closely related H5N1 viruses followed shortly in January 2021 in Mali, Nigeria and Niger [10]. 

The index case of clade 2.3.4.4 H5N1 HPAI in South Africa was an outbreak in commercial layers near Brakpan, Ekurhuleni district, Gauteng province on 9 April 2021. Ten days later the second outbreak was reported, this time in broiler breeders near Potchefstroom, North West province, and by May 2021 the disease had spread to poultry operations and wild birds in the Western Cape, Mpumalanga and Free State provinces. The first case reported in Lesotho, which is a landlocked country within South Africa’s borders, was in layer hens at Ha Penapena, Maseru district on 28 May 2021. The hens had been purchased at point-of-lay ten days earlier from a South African supplier near Parys, Free State province, and on 1 June 2021, a second outbreak in Lesotho was reported on a farm located 120 km away near Mahobong, Leribe district. These layers had been purchased from the same South African supplier which also experienced an outbreak shortly after the sales, and because of the epidemiological links, a farm-to-farm spread of H5N1 HPAI from South Africa to Lesotho was presumed [11].

Around the same time in early June, Botswana veterinary authorities started reporting mass mortalities of poultry and wild birds that lasted until September 2021. Backyard chickens, African mourning doves (*Streptopelia decipiens*) and an African fish eagle (*Haliaeetus vocifer*) were affected in the Bokaa, Kgatleng District, and/or around the Okavango Delta, Northwest District [12]. Later that year in December, Namibian veterinary authorities reported an H5N1 HPAI outbreak that caused the deaths of >6500 Cape cormorants (*Phalacrocorax capensis*) in the breeding colony at Bird Island, Walvis Bay [13]. With the limited number of sequences available at the time from Lesotho, Botswana and Namibia but none from South Africa, tentative phylogenetic links between the southern African outbreaks were made, with origins in West Africa and Europe from early 2021 [11,12,13]. In the present study, we describe the epidemiology of the South African H5N1 HPAI outbreaks from April 2021 to December 2022. Genome sequencing of more than 100 diagnosed cases from commercial poultry, wild and captive birds was performed, and phylogenetic analysis was applied to investigate the genetic relationships and spread of the clade 2.2.3.4B H5N1 HPAI virus in southern Africa. 

## 2. Materials and Methods

### 2.1. Regional and Global H5Nx Outbreak Data

A total of 10,900 individual outbreaks of H5N1 HPAI (from 320 distinct HPAI outbreak events) were extracted from the World Animal Health Information System (WAHIS) filtering for either high pathogenicity avian influenza viruses (poultry) (Inf. with) (*n* = 4945 (45.4%)) or Influenza A viruses of high pathogenicity (Inf. with) (non-poultry including wild birds) (2017-) (*n* = 5955 (54.6%)). Data were extracted on 3 March 2023 and outbreaks were included if they started between 1 July 2020 and 14 February 2023. H5N1 HPAI outbreaks were categorized by regions made up of Africa (*n* = 547 (5%)), the Americas (*n* = 1478 (13.6%)), Asia (*n* = 976 (9%)) and Europe (*n* = 7899 (72.5%)). A total of 83 countries were represented. For the H5N1 and H5N8 comparative analysis, a more detailed dataset was extracted from WAHIS of the H5N8 and H5N1 epizootic events in South Africa (2017 and 2021, respectively) for both wild bird and domestic bird populations. In this dataset, the affected population was also retrieved. This dataset consisted of 218 (56.5%) H5N8 events and 168 (43.5%) H5N1 events. Graphs were created in R [14] using the ggplot2 package [15]. Maps of HPAI outbreaks were produced using qGIS [16].

### 2.2. Field Samples

Oropharyngeal, tracheal, cloacal and organ swabs or tissues were collected by veterinarians, animal health technicians and researchers from suspected clinical cases in sick and dead wild birds and poultry, across South Africa between April 2021 and December 2022. Samples were collected as part of passive government disease surveillance, active surveillance in African penguins, environmental fecal sampling of wild ducks [17], and a disease surveillance project at a seabird rehabilitation center. Active surveillance and rehabilitation center samples were collected into viral transport medium (VTM) comprising brain heart infusion (BHI) broth (pH 7.2) (Oxoid Ltd., Basingstoke, UK), 10% (*v*/*v*) glycerol and the following antimicrobials per liter: 100 mg doxycycline (Mylan, Potters Bar, UK), 100 mg enrofloxacin (Cipla, Beckenham, UK), 1000 mg penicillin-streptomycin (Sigma-Aldrich, Merck, Sofia, Bulgaria), and 5 mg amphotericin B (Bristol–Myers Squibb, Denham, UK).

The hosts included commercial or backyard chickens (*Gallus gallus domesticus*) (*n* = 52), commercial ostriches (*Struthio camelus*) (*n* = 5) as well as captive and free-living wild birds (*n* = 54) in the Gauteng, North West, Mpumalanga, Free State, KwaZulu-Natal and Western Cape provinces. The wild bird species included waterfowl: Egyptian goose (*Alopochen aegyptiacus*)*,* Sacred ibis (*Threskiornis aethiopicus*), European white stork (*Ciconia ciconia*), black-headed heron (*Ardea melanocephala*), blue crane (*Grus paradisea*); as well as raptors: spotted eagle-owl (*Bubo africanus*), African barn owl (*Tyto capensis*), African fish eagle; and seabirds: Hartlaub’s gull (*Chroicocephalus hartlaubii*), African penguin, kelp gull (*Larus dominicanus*), a great white pelican, Cape cormorant, Cape gannet (*Morus capensis*), white-breasted cormorant (*Phalacrocorax lucidus*), northern giant petrel (*Macronectes halli*), bank cormorant (*Phalacrocorax neglectus*), common tern (*Sterna hirundo*), swift tern and African black oystercatcher (*Haematopus moquini*). 

The samples were submitted to government-accredited veterinary laboratories for molecular diagnosis, where H5-subtype IAV (Influenza A virus) infection was confirmed using the real-time RT-PCR methods recommended by the European Union Reference Laboratory (https://www.izsvenezie.com/reference-laboratories/avian-influenza-newcastle-disease/diagnostic-protocols; accessed on 14 June 2023). Extracted total nucleic acids or RNA from positive cases were forwarded on ice packs to the University of Pretoria for sequencing. This study aimed to sequence the genome of at least one representative virus from each reported outbreak, but samples from some backyard poultry and wild bird outbreaks were unavailable for analysis. Tissue samples from five suspected H5N1 cases in commercial poultry (UP01/2021, UP481/2021, MAB/2021, UP928/2021, PRL118/2022, and PRL60/2022) were submitted directly to the Poultry Section of the University of Pretoria. High-throughput total nucleic acid extraction and real-time RT-PCR detection of IAV in samples submitted to the University were performed as described previously [17]. 

### 2.3. Genome Sequencing and Assembly

IAV genomes were amplified in single-tube RT-PCR reactions and sent to the University of Stellenbosch Central Analytical Facility for Ion Torrent sequencing as previously described [17]. Ion Torrent reads were imported into the CLC Genomics Workbench 22 (QIAGEN CLC bio, Aarhus, Denmark) and assembled to reference sequences representing the eight segments encoding the IAV hemagglutinin (HA) subtypes H1 to H16 (segment 4), neuraminidase (NA) N1 to N9 (segment 6), plus those encoding the internal genes. Some genome segments encode more than one gene [18] but for convenience, they are referred to here as PB2 (segment 1; polymerase B2); PB1 (segment 2; polymerase B1); PA (segment 3; polymerase A plus PA-X); NP (segment 5; nucleocapsid); M (segment 7; matrix 1 plus matrix 2e); and NS (segment 8; non-structural 1 plus nuclear export protein) (GenBank accession numbers KY621531-KY621538, JN696316, KF313565, MH412114, GU122032, MH637350, KM054845, MH637361, HE802739, KT777901, KP087869, HE802715, KX101133, MH637340, KP287772, MH637353, KM244048, MH411954, CY080155, KM244094, MH637406, KM244102, MH637420 and KJ484622). The consensus sequences for full and partial H5N1 genomes generated in this study are available in the GISAID EpiFlu database under accession numbers EPI_ISL_145424488-EPI_ISL_17072111. 

### 2.4. Sequence Analysis

Closely related reference sequences for each of the eight IAV genome segments were retrieved from the GISAID EpiFlu and GenBank (https://www.ncbi.nlm.nih.gov/nucore; accessed on 11 April 2023) databases by BLAST analysis. After removing duplicates, the top 100 matches in each were used for the initial analysis, but ultimately this was downsampled to the closest relatives only. Full or partial sequences were available for the H5N1 outbreaks in Lesotho (*n* = 2; [11]), Botswana (*n* = 6; [12]) and Namibia (*n* = 1; [13]). MEGA-X (v10.2.5) [19] was used to concatenate the complete genome sequences and subsequently generate a distance matrix. The distance matrix, which estimates the evolutionary divergence between sequences, was generated by the maximum composite likelihood method with results expressed as the number of nucleotide substitutions per site. Multiple sequence alignments (MSA) of the eight individual segments and the concatenated genomes were prepared using the MAFFT v.7 (https://mafft.cbrc.jp/alignment/server/index.html; accessed on 11 April 2023 [20]) and BioEdit [21], with protein sequence translation in BioEdit. Phylogenies were reconstructed using maximum likelihood (ML) analysis in IQ-Tree v2.0.3 with 1000 ultrafast bootstrap replicates [22], and the ML consensus trees were visualized using FigTree 1.4.4 (http://tree.bio.ac.uk/software/figtree/; accessed on 12 April 2023). For reassortment analysis, the radial views of the ML trees were examined and sub-clades were identified and defined based on the statistical support for the nodes (bootstrap values). 

The time to the most recent common ancestor (tMRCA) of the HA gene was inferred from the dated maximum clade credibility (MCC) tree in BEAST v.2 software [23]. The MCC tree was reconstructed using a Hasegawa–Kishono–Yano (HKY) nucleotide substitution model with a gamma distribution of substitution rates, a Coalescent Bayesian Skyline model and a relaxed lognormal clock. Markov chain Monte Carlo chains of 50 million iterations were performed and assessed with Tracer v1.7.2 [24] to ensure that an effective sample size (ESS) of >200 was achieved, with a statistical uncertainty of the nodes reflected in values of the 95% highest posterior density (HPD). The consensus MCC tree with common ancestor heights was summarized in TreeAnnotator v.2.6.6 and visualized using FigTree v.1.4.2 (UK). Sequence names in phylogenetic trees were shortened using the code “ZA” for South Africa. 

## 3. Results and Discussion

### 3.1. Epidemiology of Clade 2.3.4.4B in South Africa

Figure 1 and Figure 2 represent the H5N1 reports to WOAH between July 2020 and February 2023. In Figure 1 all non-southern African country outbreaks are depicted in grey with the four southern African countries affected (South Africa, Botswana, Lesotho and Namibia) indicated separately with their own color. The South African H5N1 epizootic was preceded by relatively small numbers of cases from Asia and started at approximately the same time as the European outbreaks, although other African countries had cases before those in Europe. The majority of cases had already occurred in South Africa and Africa by the time the Americas, Asian and European outbreaks increased substantially. However, the H5N1 strains that caused the outbreaks in the Americas, Asia and Europe were phylogenetically distinct from those affecting southern Africa, and only the closest relatives were included as references in the final phylogenetic trees analyzed in this study (Appendix A). 

The global extent of these outbreaks is shown in Figure 2. Outbreaks in Africa are very distinctly clustered in western and southern Africa with outliers reported from Reunion Island and northern Africa in fewer numbers. Southern Africa is inset into the image to depict the cases in South Africa, Lesotho, Botswana and Namibia more clearly.

A comparison between the 2017/2018 H5N8 and 2021/2022 H5N1 epizootics was undertaken. Both epizootics had a distinct bi-phasal epidemic curve with a similar pattern which fits with the propagating epidemic pattern of avian influenza. Evaluation of epizootic by month of occurrence (Appendix A) shows the H5N8 epizootic initial (3 months into epizootic) and secondary peaks (9 months into epizootic) preceding those of the H5N1 epizootic (7 and 12 months into epizootics, respectively). The broad classification of wild or domestic species affected was similar with a wild: domestic ratio of 47.7:52.4 for H5N8 and 43.5:56.5 for H5N1. There were differences, however, at a more specific species level. While the data quality is difficult to classify across all outbreaks, the one species that is readily classifiable is that of ostriches. Ostriches were associated with 22% of all outbreaks, but in the H5N8 epidemic, they made up 32.5% of outbreaks and 62.3% of domestic bird outbreaks. In the H5N1 epidemic, they make up 8.3% of all outbreaks and 14% of domestic bird outbreaks. Spatially, the distribution of outbreaks was similar across provinces; the Western Cape province had the most outbreaks in both H5N8 (75.7%) and H5N1 (45.8%) events. The difference in proportions between the two epidemics was made up of the Gauteng and Kwa-Zulu Natal (KZN) provinces, with increases from H5N8 to H5N1 being 0.5% to 13.7% for KZN and 6.9% to 27.4% for Gauteng.

### 3.2. Western African Origins of the Southern African H5N1 HPAI Viruses

Complete (*n* = 97) or partial (*n* = 20) South African clade 2.3.4.4 H5N1 HPAI virus genome sequences were generated and compared with those available from outbreaks in Lesotho, Botswana and Namibia, and other close relatives retrieved in sequence homology searches. A notable feature in the eight individual ML trees (Appendix A) and the concatenated genome ML tree (Figure 3) was the clustering of all South African, Lesotho and Namibian viruses from 2021–2022 into a single monophyletic clade except for a single South African virus, ostrich/PRL060/2022 that grouped separately with Botswana’s viruses from 2021. 

The ML trees concur with the findings of previous phylogenetic studies [11,12,13] that the clade 2.3.4.4B H5N1 strains in West Africa in late 2020 and early 2021, but especially Senegal, were the progenitors of the southern African viruses. The molecular clock analysis of the HA gene (Figure 4) dated the time to the most recent common ancestor (tMRCA) of the general South Africa/Lesotho/Namibia sub-clade to December 2020 (95% HPD October 2020–February 2021), consistent with the timing of the Senegalese outbreaks first detected on 23 December 2020 [9]. The tMRCA of the outlier South Africa/Botswana sub-clade was dated somewhat later in May 2021 (95% HPD March–June 2021). The South Africa/Botswana sub-clade had no recent close ancestors but shared an earlier recent common ancestor (RCA) with H5N1 HPAI viruses from the Russian Federation, the Czech Republic, Egypt and Nigeria in 2021, which is dated September 2020 (95% HPD July–December 2020). The long branch length of ostrich/PRL060/2022 suggests a prolonged sub-regional circulation prior to its introduction to the Free State province in late 2022. No reassortment with African low pathogenic IAV subtypes was detected. 

### 3.3. Introduced and Gained Genetic Diversity

The concatenated full genome ML tree (Figure 3) provides a useful overview of the total genetic distances and phylogenetic relationships between the southern African viruses, but in order to investigate the fine-scale epidemiological spread of H5N1 HPAI within the sub-region it was necessary to evaluate genomic reassortments between the viruses, including those for which only partial genomes were obtained. To achieve this, the radial views of the individual ML trees were examined and sub-clades were identified and assigned a unique letter and number (Appendix A). The sub-clades in the M and NS genes were less obvious and had low bootstrap values; therefore, only two sub-clades each were assigned here. Ultimately, sub-clades A1–4 (PB2), B1–4 (PB1), C1–3 (PA), D1–4 (HA), E1–4 (NA), G1 & 2 (M) and H1 & 2 (NS) were assigned, tabulated, and are presented for comparison in Figure 5, where the combination of sub-clades designates a sub-genotype. Thus, fifteen southern African (SA) sub-genotypes were identified that are also fully supported by the topology and bootstrap values > 90% in Figure 3. The South Africa/Lesotho/Namibia sub-cluster is represented by sub-genotypes SA1 to SA14 and the unique SA/Botswana sub-cluster was designated as sub-genotype SA15. The insights gained from the genomic reassortment analysis read in conjunction with the temporal distributions (Figure 6) and mapped geographic locations (Figure 7) are discussed below.

Seven distinct H5N1 HPAI sub-genotypes (SA1 to SA5, SA7 and SA8) were detected within the first six weeks of the outbreaks in South Africa in April and May 2021 (Figure 6 and Figure 7), and this high genetic diversity suggests that multiple relatively heterogenous viruses were introduced almost simultaneously. The index case in commercial layer hens near Brakpan, Gauteng province on 9 April 2021 (26683/21) is an SA1 strain, whereas the second outbreak on 19 April in Potchefstroom, North West province (26700/21) is an SA3 strain. Sub-genotypes SA1 and SA2 were only found in the north-central regions of South Africa and Lesotho (Figure 7) and were not detected again after early June 2021, whereas SA3 (*n* = 12), SA4 (*n* = 16) and SA5 (*n* = 31) viruses were found in both the north-central regions and much further south in the Western Cape province in the first 6 weeks of the epizootic. SA3 circulated in both the northern and southern regions until August 2021, and an SA3 virus was likely the progenitor of SA6, which differs by a PB1 reassortment and is represented by a single virus (692881/21) that caused an outbreak in commercial chickens near Wellington in the Western Cape in mid-May 2021. 

SA4 was found almost exclusively in the Western Cape province where it caused outbreaks in commercial chickens, coastal seabirds and other wild birds from May to November 2021. The only exception was 695344/2021 from an outbreak in commercial chickens in the Pretoria region, Gauteng province on 8 June. However, since only a partial genome could be sequenced the designation of 695344/2021 as SA4 is tentative. SA8, with a sole representative in African barn owl/21050429/21, was the result of reassortment between an SA4 virus (from which it differs by a double HA and NA reassortment) and SA10, with which it shares the most closely related HA and NA genes. 

SA5 was predominant in the Western Cape province from early May 2021 until February 2022 where it caused multiple outbreaks in commercial birds, especially layer hens, but it was also detected in commercial ostriches as well as a wide variety of wild birds including Egyptian geese, Hartlaub’s gulls, Sacred ibis, an African fish eagle, blue cranes and Cape gannets. In late June 2021, SA5 appeared in Ashburton, KZN province where it caused an outbreak in commercial chickens (697352/2021), and in August 2021 in an outbreak in 68-week-old commercial layers in Pretoria, Gauteng province (10766/21). The virus may have persisted in the north-central region until at least October 2021 when it was detected during active surveillance of wild ducks in the Bronkhorstspruit region (GDARD TF04/2021; [17]), but since only partial genomes were available for the latter two cases their classification as SA5 is tentative. Sub-genotypes SA7 and SA9 are represented by single viruses that are related to SA5; SA7 differs from SA5 in a PA reassortment, and the sole representative, UP196/21, was collected during active surveillance of wild ducks near Standerton, Mpumalanga province in mid-February 2022. SA9 (2107586/2021) was detected in ostriches near Touwsrivier, Western Cape province differing from SA5 in the PA gene, but other unique features of this virus are discussed later on. 

The ancestral nodes of these early sub-genotypes in South Africa (SA1, SA2, SA3/SA6, SA4/SA8 and SA5/SA7/SA9) pre-date the 9 April index case, but the posterior probabilities in the HA gene MCC tree are generally low (Figure 4). Therefore, precise dating of when they arose was not possible, but their progenitor emerged sometime between late December 2020 and March 2021. The sub-genotypes detected after May 2021, viz. SA10 to SA14, likely emerged within South Africa and are discussed along with SA15 in further detail below. 

### 3.4. South African Poultry Was Not the Source of the Lesotho Outbreaks

To recap the case history: in June 2021 a South African supplier of point-of-lay hens sold birds to producers in Lesotho, and within a short space of time all three farms were affected by H5N1 HPAI outbreaks, which led the veterinary authorities to conclude at the time that the outbreak spread with the movement of infected birds from South Africa. The molecular data presented here, however, shows that this was not the case. Most notably, the two Lesotho outbreaks were caused by sub-genotype SA2 strains, whereas strain chicken 693965/2021 from the Free State supplier’s outbreak was classified as sub-genotype SA3. A third virus from South Africa, 690841/21, from a commercial chicken outbreak in Standerton two weeks prior to Lesotho’s outbreaks, was also tentatively classified as SA2, but only the HA and NA genes were available. Coincidentally, the same Free State supplier also sold hens to a producer in Pretoria, Gauteng, where an outbreak started on 8 June 2021, represented by chicken/695344/2021. Again, it was speculated that farm-to-farm spread had occurred, but our results show that this partially sequenced virus is classified as SA4. Nonetheless, it seems that Lesotho’s outbreaks were caused by its own unique sub-genotype (SA2). In Figure 3, the Lesotho viruses form an outgroup to the majority of SA’s strains with strong statistical support (bootstrap value of 92%). Additionally, in the HA MCC tree (Figure 4), the tMRCA of the Lesotho viruses with others from SA was dated January 2021 (HPD Nov 2020-Apr 2021). This, then, pre-dates the South African index case, although the posterior probability is too low to state this definitively. Therefore, it is most likely that SA2 was introduced to the sub-region alongside the other “early” sub-genotypes and circulated for some weeks in wild birds in Lesotho prior to the spillover in chickens in that country.

### 3.5. Outbreaks of H5N1 HPAI in Commercial Ostriches Were Point Introductions

Commercial ostriches are regularly screened to detect exposure to avian influenza using serological tests, and the detection of any influenza A antibodies triggers swab collection for virus detection. Under experimental conditions, ostriches infected with clade 2.3.4.4B H5 HPAI may appear clinically healthy whilst shedding high amounts of virus, but if they ingest an excessive viral load from the environment, or if they are stressed and immune-compromised, they can develop typical HPAI-related neurological and respiratory signs and die within a few days [25]. Overall, there were fewer reported cases of H5N1 HPAI in commercial ostriches in 2021/2022 compared to the 2017/2018 H5N8 outbreaks, with only five ostrich-origin viral genomes available for analysis this time.

A sub-genotype SA5 strain, 21060357/2021, was recovered on 17 June 2021 from an outbreak in 8-month-old commercial ostriches near Swellendam, Western Cape. At least a third of the flock died in this outbreak, with congested organs seen in one bird post-mortem. This virus was genetically most closely related to UP197/21 (SA7), detected seven months later and 1440 km away in environmental wild duck fecal swabs collected at a dam near Standerton, Mpumalanga province (Figure 3). Then, 21060425/2021, detected on the same ostrich farm six days later, and 21060311/2021, from an outbreak near Albertinia, were similarly classified as the SA5 sub-genotype that predominated in the Western Cape at the time. Although these two ostrich viruses are genetically similar across the full genome (0.0001 nucleotide substitutions per site; Appendix A), the farms were approximately 75 km apart with no known epidemiological links. Notably, 21060311 was identical in the consensus genome sequence to those of five other H5N1 viruses from the Western Cape in May and June in commercial and backyard chickens and wild blue cranes that succumbed to the disease (blue crane/21060475/2021, chicken/21050119/2021, chicken/21050125/2021, chicken/21060265/2021 and chicken/21060469/2021; see Figure 3; Appendix A), pointing to the wide dissemination of this particular strain in the Western Cape’s wild birds and poultry sector preceding and coinciding with these ostrich outbreaks.

Next, 21070586/2021, a unique sub-genotype SA9 strain, was detected in 5–8-month-old ostriches (grower phase) on 28 July 2021 on a farm near Touwsrivier, Western Cape. A mortality rate of approximately 15% was recorded in June and July and samples taken post-mortem on 12 July had tested HP H5 rRT-PCR-positive. Affected birds appeared to be in good body condition but showed clinical signs including depression, recumbency, anorexia and sudden death within 24 h. Other clinical signs included neon-green feces, apparent sinusitis with purulent ocular discharge and severe ulcerative stomatitis. Post-mortem signs included petechial hemorrhage on the serosal surfaces of the small intestines, in the trachea and on the epicardium. The farm housed multi-age ostriches (breeders, chicks and slaughter birds) and Touwsrivier, located in the driest part of the Klein Karoo semi-desert, is a veritable oasis attracting wild birds to its irrigated lucerne pastures and dams. Blood tests from the grower bird epidemiological group (7 weeks to slaughter age) sampled on 9 June 2021 were AI-antibody negative, but by 19 July, serological testing indicated that approximately 54.4% of the grower flock had seroconverted, most with clade 2.3.4.4 H5-specific antibody hemagglutination inhibition titers reaching 1:256 (A. Olivier, pers. comm.). The primary virus introduction could have occurred any time from mid-May 2021, and, notably, this virus contained an E627K mutation in the PB2 protein, which is associated with mammalian adaptation [26], but is also a known marker of ostrich adaptation that emerges after a period of transmission within a flock [27]. 

Interestingly, PRL060/2022, the SA15 virus from the outbreak near Fauresmith, Free State province in November 2022 that shared an RCA with the Botswanan strains, also contained the ostrich-specific E627K mutation in the PB2-protein (but E627K was absent in three PB2 sequences available from Botswana). The virus was similarly detected in 6–8-week-old ostrich chicks, but these birds displayed classical neurological signs including incoordination, “star gazing” and sudden collapse when excited. The birds preferred to be recumbent, and when disturbed seemed to be blind, disorientated and uncoordinated. Good rains in the higher-lying Orania/Hopetown region transformed what is normally a dry catchment area at the bottom of a valley where Fauresmith is situated into a wetland for large parts of the spring season. Large numbers of waders and other smaller wetland birds as well as larger species such as Sacred ibises were observed there, and passerine bridging hosts are suspected to have introduced the virus as fomites into the chick camps (A. Olivier, pers. comm.). The HA tMRCA of the SA/Botswana sub-clade was dated to early May 2021 (95% HPD Feb–June 2021) (Figure 4) but no SA15 or similar viruses were detected in South Africa throughout 2021 or most of 2022, and therefore this sub-genotype was likely a new introduction into the region just weeks before its detection in the ostriches, and the timing is consistent with a southward spring migration of the unknown wild hosts.

### 3.6. Limited Farm-to-Farm Spread in South African Commercial Chicken Outbreaks

Despite their intensified efforts to maintain strict biosecurity, poultry producers were heavily affected by H5N1 HPAI outbreaks in 2021, with sporadic cases in commercial, small-scale and backyard chickens continuing into the first month of 2022. IAV whole genome data was essential to determine whether outbreaks were due to point introductions (a localized biosecurity break where the external environment was heavily contaminated by wild bird-origin H5N1 viruses) or were caused by farm-to-farm spread by contaminated vehicles, people, feed and equipment. In many cases, the genetic data was unequivocal that outbreaks in commercial chickens were caused by point introductions. For example, in the Western Cape province between the 6 and 10 May 2021, almost simultaneous outbreaks occurred in a single large producer’s operations near Malmesbury (21050090/2021) and Worcester (21050118/2021, 21050119/2021, 21050125/2021 and 21050021/2021). Thus, 21050090/2021 (Malmesbury; 6 May) was phylogenetically most closely related to coastal seabird viruses kelp gull/21050387/2021 and Hartlaub’s gull/21050295/2021 (all sub-genotype SA4) (Figure 3), whereas all the cases near Worcester in that same week were caused by SA5 strains. Among the latter, 21050118/2021 was phylogenetically distinct from the other three outbreaks in this producer and shared an RCA with 693331/2021 (an outbreak in layers near Wellington on 23 May, from a different producer), 236744/2021 (the layers of yet another producer near Paarl on 19 October), and a Cape gannet virus (21100108/2021) sampled at Lambert’s Bay in early October. The affected farm is 10 km south of the other three outbreaks, which were on different sites of the same farm.

Other sub-genotype SA5 viruses in the Western Cape that caused chicken outbreaks grouped most closely with wild bird viruses. Indeed, 693781/2021 caused an outbreak in 14-week-old layers near Malmesbury on 27 May, and its closest relative was detected in a dead pelican (21050494/2021) found two days later in the same area, and 692329/2021, from an outbreak near Grabouw in mid-May, shared a most recent common ancestor with an African fish eagle virus (21060065/2021) diagnosed in a weak bird found near Bredasdorp at the end of June. Later on, 21060435/2021 was associated with an outbreak causing 100% mortality in backyard chickens in Phillipi, Cape Town on 24 June, but this is an SA3 strain. Next, 21050299/2021, the cause of the 18 May outbreak near Piketberg, an SA4-type virus, was genetically distinct from all other chicken outbreaks in the Western Cape at the time and shared ancestry with what would eventually emerge as the SA13 coastal seabird-specific lineage that is discussed in the next section. Viruses from outbreaks in other provinces were also genetically distinct, and therefore point introductions into those operations. Examples include UP481/2021 (SA1, broiler outbreak in Villiers, Free State province on 10 May) and MAB/2021 (SA3, Rietvlei near Pretoria, Gauteng province on 27 August).

To evaluate possible secondary spread in cases within the same sub-genotype where the genetic distances were smaller, the phylogenetic grouping in the concatenated genome tree (Figure 3) was used in conjunction with the distance matrix (Appendix A). In the distance matrix, we considered a cutoff value of ≤0.001 nucleotide substitutions per site between two sequences to be a possible secondary spread, unless closely related or identical wild bird viruses were located in the same sub-clade as the chicken virus (a point introduction cannot be ruled out). Outbreaks in poultry in geographically separated regions with no known epidemiological links were also disregarded.

Of the 44 chicken-origin complete genomes available for analysis, in three cases the sequenced viruses were sampled from the same farm on the same day. In addition, 21080238/2021 and 424469/2021 were sampled at the same time from an outbreak in a single free-range layer producer’s layer hen operation near Stellenbosch but tested at different laboratories. An interesting case was an outbreak in broiler breeders in Worcester, Western Cape province in early May 2021: 21050021/2021 and 690575/2021 were sampled from different houses on the same farm, on the same day, but 690575/2021 was located on a relatively long branch in the phylogenetic tree (Figure 3), and the distance between these two viruses was 0.0011 nucleotide substitutions per site. Such genetic distance indicates the diversity of viruses in the external natural environment at the time the biosecurity breach occurred. In contrast, 22010387/2022 and QF22304/2022, also sampled from a single outbreak one day apart (layer hens, Wellington, late January 2022), were highly similar across the genome (0.0002 nucleotide substitutions per site). QF22465/2022 was sampled from a different house on the same site one week later, with 0.0008 nucleotide substitutions per site (Appendix A) and, therefore, the second outbreak was likely a secondary spread within the producer’s operations.

Sub-genotypes SA10, SA11 and SA12 form a distinct and strongly supported sub-clade in Figure 3, comprising viruses from a cluster of outbreaks in the Kwa-Zulu-Natal province between the end of June to early September 2021, and that later on appeared in Gauteng. The MCC HA gene analysis (Figure 4) dated the RCA of sub-clades SA10/SA11/SA12 to May 2021 (95% HPD April to June), i.e., after the index cases, signifying that the ancestral SA10/11/12 virus probably emerged within South Africa’s borders. The only SA10 virus from the Western Cape province, 21100506/2021, from an outbreak in layer hens near Yzerfontein at the end of October, is a tentative assignment to SA10 because only the HA and NA genes were available for analysis (it is equally likely to be a unique sub-genotype). The partial genome availability was due to an H9N2 co-infecting virus. The closest relatives (>97% nucleotide sequence identity) to the partial H9 and N2-specific genes as determined by BLAST analysis were the Eurasian-type H9N2 strains A/Mallard(*Anas platyrhynchos*)/South Korea/KNU2021-41/2021(H9N2) (accession number ON505892) and A/Bean Goose(*Anser fabalis*)/South Korea/KNU 2019-16/2019(H9N2) (accession number MW380632), respectively, providing supporting evidence that the Yzerfontein outbreak was caused by a point introduction from wild birds. Nonetheless, the homology in the H5N1 HA and NA genes suggests an epidemiological link between the Western Cape and KwaZulu-Natal viruses, and that wild birds may have spread the virus north-eastwards, potentially driven from the southern Cape region by an extreme cold front in June 2021.

The Camperdown region of the KZN province has a high poultry density, dominated by a single large producer that operates multiple broiler breeder and layer hen sites there. The outbreaks in the Camperdown region started in late June 2021 and lasted until early September 2021. Since the large producer was most affected and outbreaks occurred within a relatively short space of time, the farm-to-farm spread was assumed, but the molecular evidence shows otherwise. Firstly, SA11 (412374/2021), which is an NP reassortant, can unequivocally be identified as a point introduction. A sub-cluster comprising 33069/2021, 33071/2021, 411542/2021, 411258/2021, 412374/2021 and 412364/2021, all sampled between 20 to 30 July 2021, cannot be ruled out as farm-to-farm spread because of their close phylogenetic relationships (Figure 3). However, as a cluster supported by a 99% bootstrap value, they collectively represent a second point introduction. Then, 412380/2021 and 412381/2021 were isolated from the same site on consecutive days during December 2021; the viruses are not identical (0.0006 nucleotide substitutions per site; Appendix A) and their location on a separate branch with 100% support, plus 0.0011–0.0032 nucleotide substitution differences with other viruses in the general cluster, indicates a third primary introduction to the region. To be sure, 411255/2021 contains sufficient changes across the genome (0.0015 nucleotide substitutions per site) to designate it as a fourth-point introduction; 411262/2021 and 683320/2021 are phylogenetically basal to the others and cannot be separated or designated as a unique introduction but 683320/2021 caused an outbreak in the hens of a different producer; 33081/2021 and 412372/2021 (sub-clade supported by a high bootstrap value) had a distance within the cutoff of 0.0009 nucleotide substitutions per site. These two SA10 viruses were collected from simultaneous outbreaks in different producers, located 50 km apart in Howick and Camperdown, respectively, with no known epidemiological link, and were therefore not considered a secondary spread event and represent the fifth-point introduction to KZN. The ecological factors causing the high levels of H5N1 environmental contamination at the time in the KwaZulu-Natal province remain unknown, and no large die-offs in wild birds were reported here. 

The SA10/SA11 sub-cluster from KZN and the SA10/SA12 sub-cluster detected later in Gauteng shared an RCA that was dated June 2021 (95% HPD May–Aug 2021) (Figure 3). The first SA10-associated outbreaks in poultry in the Gauteng province are represented by DW2201/2022 (Kempton Park) and JB2201/2022 (Elandsfontein), in simultaneous outbreaks in late January 2022 in the layers of different producers located 32 km apart. Although the genetic distance between these viruses was only 0.0002 nucleotide substitutions per site, epidemiological tracing could not establish a link between these farms. Subsequently, PRL118/2022 was the cause of an outbreak in “speculator” chickens near Fochville on 20 February 2022, but the significant genetic distance (Figure 3) as well as the NA reassortment that designates this virus as a unique SA12 sub-genotype rules out the Kempton Park and Elandsfontein outbreaks as the source. The most recent SA10 virus sequenced, 697683/2022, was recovered from an outbreak in 31-week-old commercial layers in Cato Ridge, KwaZulu-Natal province in mid-September 2022. Its closest relative, however, was BA107/2022, sampled from a sick European white stork caught during active wild bird surveillance at Bon Accord dam near Pretoria at the beginning of February 2022 [4]. The phylogenetic data suggest SA10/SA12-like viruses persisted in the Gauteng province wild bird reservoir until late 2022, which then spread the virus back to KwaZulu-Natal because the Cato Ridge strain was evidently unrelated to the 2021 outbreaks in KwaZulu-Natal. 

### 3.7. H5N1 HPAI in Raptors

Raptors become infected during H5N1 HPAI outbreaks presumably through predating on infected sick prey or carrion. In addition to the African fish eagle case near Bredasdorp mentioned above (21060065/2021; SA5), a bird park in Hout Bay, Western Cape province experienced an outbreak between 19 May and approximately the end of July 2021, which is represented by a discrete sub-clade of SA5 viruses (Figure 3). HPAI was blamed for 152 deaths recorded between 19 May and 20 July. At least seven spotted eagle-owls, nine black-headed herons and four African barn owls perished, along with smaller numbers of at least eight other raptor species, and two other species of heron, turkeys and ducks that were part of the collection. Deaths recorded in Hadeda ibises (*Bostrychia hagedash*), African rock pigeons (*Columba guinea*), laughing doves (*Spilopelia senegalensis*) and gulls are assumed to be either rescued birds brought to the park by the public, or birds found in open displays in the park. The spotted eagle-owl virus (21050363/2021) from 19 May was very closely related to a black-headed heron virus (21050430/2021; SA5) from 26 May in the same facility (0.0001 nucleotide substitutions per site; Appendix A). The African barn owl virus (21050429/2021), also from 26 May, contained slightly greater genetic distances from these (0.0007–0.008 nucleotide substitutions per site). However, a Buff Orpington chicken in the collection, sampled on 19 May, succumbed to an SA3 strain (21050364/2021).

### 3.8. Emergence of a Coastal Seabird-Specific H5N1 Sub-Lineage in South Africa That Spread to Namibia

H5N1 HPAI appeared relatively early in Western Cape coastal seabirds compared with the H5N8 HPAI epizootic, and at least two distinct sub-genotypes were co-circulating in the gull populations early on. Hartlaub’s gulls were among the first species affected in and around Cape Town, with the first case found on 14 May 2021. SA3 virus strains were identified in three Hartlaub’s gulls (21050388/2021, 21050385/2021 and 21050384/2021), a kelp gull (21050387/2021) and an African penguin (21050383/2021). These are genetically similar to the other SA3/SA6 strains that were being diagnosed in poultry and terrestrial wild birds in outbreaks in the Western Cape and more northern provinces in a similar period (Figure 3, Figure 5 and Figure 7). Hartlaub’s gull/21050295/2021 and kelp gull/21050299/2021, also from around Cape Town, were, however, identified as SA4 strains. SA5 viruses also spilled into coastal seabirds at some point. One of 40 dead pelicans (21050494/2021) was diagnosed from a farm dam near Malmesbury at the end of May; the only mass mortality recorded in pelicans and SA5 also caused mortalities in Cape cormorants (21100108/2021), and presumably in Cape gannets, in Lambert’s Bay in October.

A distinct cluster of SA4 viruses in coastal seabirds in Figure 3, supported by a high bootstrap value, would ultimately give rise to the seabird-restricted sub-lineage of viruses, SA13. Basal to this cluster of ancestral SA4 viruses is cormorant/21120147/2021 (Nature’s Valley Beach, 6 December), the sole representative of SA14, which differs from SA13 by a PB1 reassortment (Figure 5). The cluster of progenitor viruses, sampled from mid-October to November 2021, comprise a white-breasted cormorant (21100215/2021), a bank cormorant (21110253/2021), a northern giant petrel (21100283/2021) and viruses in African penguins (21100423A/2021, 21100423B, 21100423C and 21100423D) on Dyer Island. The latter viruses were associated with mass mortality in both African penguins (at least 200 deaths) and, presumably, Cape cormorants (at least 15,000 deaths) at the same time, during the peak Cape cormorant breeding season. The viruses were all sampled from the Dyer Island colony on 26 October but, interestingly, they are not identical. Most notably, 21100423D/2021 was homologous with the northern giant petrel virus sampled 10 days earlier at St Helena Bay, 320 km away. The genetic distance between this cluster of African penguin H5N1 viruses ranged from 0.002 to 0.011 nucleotide substitutions per site, with the high genetic diversity indicating the likelihood of multiple-point introductions to the island. 

Chronologically the next cluster of related viruses, this time detected predominantly in Cape cormorants, represents the emergence of the seabird-restricted sub-genotype SA13. SA4 and SA13 differ by an NP gene reassortment, and the specific NP gene of SA14 was unique to the coastal seabird lineage. The variation in the NP gene was limited to the nucleotide sequence level, which is expected since the nucleocapsid protein is not associated with host specificity or adaptation. None of the unique protein markers that distinguished the South African/Namibian coastal seabird H5N8 HPAI sub-lineage in 2017/2018, viz., N11K and T29S in HA, R95K in M1 and P559A in PA [5], were present in the SA4/SA13/SA14 viruses.

The first example of the newly emerged SA13 viruses was first detected in Cape cormorants in mid-September 2021 at Glencairn, Cape Town (21090179C/2021), then they appeared three weeks later in Cape Cormorants in Lambert’s Bay (21100109A and 21100109B/2021), in Betty’s Bay in mid-October (21100176B/2021) (900 dead chicks were found at this site in mid-November) and, finally, caused a kelp gull mortality in Muizenberg in early January 2022 (22010082/2022). Lambert’s Bay and Muizenberg were epidemiologically linked by highly similar viruses 21100109A/2021 and 22010082/2022. The tMRCA of the SA4/SA13 sub-clade progenitor was dated June 2021 (95% HPD May–August 2021) and the SA13 progenitor July 2021 (95% HPD June–August 2021), the latter indicating that the SA13 virus was probably already circulating in coastal seabirds for several weeks before the increased mortalities in Cape cormorants started. 

Our phylogenetic evidence shows that the Cape cormorant virus detected at Bird Island, Walvis Bay in Namibia in December 2021, shared an RCA with the viruses detected in South Africa’s Cape Cormorants from September 2021. Indeed, the RCA to the Namibian virus in Figure 4 was dated September 2021 (95% HPD Jul–Nov 2021, albeit with a low posterior probability). This evidence points towards a spread of H5N1 up the southwestern African coast to Namibia between September and December in an unknown seabird host, but no deaths in other species were reported along the Namibian coastline or in Walvis Bay at the time of the outbreak [7].

H5N1 HPAI outbreaks in coastal seabirds in South Africa continued in 2022. The molecular data established an epidemiological link between swift tern/681998 ST009/2022, sampled near Stellenbosch on 22 March 2022, and Cape cormorant/681998 DOA059/2022 from Fish Hoek on 23 March 2022, and these viruses, in turn, shared an RCA with an earlier African penguin case (a chick with no clinical signs) at Stony Point on 18 November 2021. Sporadic detections in common terns and an African black oystercatcher occurred throughout March and into June 2022, with the viruses (22060305/2022, 681998 CT010/2022, 686940 CT013/2022 and 682489/2022) exhibiting continuing genetic drift in the SA13 sub-genotype.

The most recent cluster of seabird outbreaks affected African penguins, starting with at least twenty cases on Dyer Island at the end of July 2022 (including 693215-P10347 and P10348/2022) before spreading in September to the Simon’s Town colony, with a related common tern virus sampled in the interim at Macassar Beach (693799 DOA236/2022). Viruses detected among at least 60 penguin deaths in Simon’s Town in September and October (698730 DOA308/2022, 698162 AP446/2022, 700164 DOA318/2022 and 702068 DOA370/2022) were phylogenetically related to those causing the death of a penguin in November at Stony Point (702626 DOA410/2022) and one at Strandfontein (702626 AP821/2022), both within swimming distance of Simon’s Town.

## 4. Conclusions

For the second time in less than five years, clade 2.3.4.4 H5Nx viruses were introduced into the southern African region, in epidemiologic waves that mimic events in the northern hemisphere. In 2017/2018 Zimbabwe, South Africa and Namibia were affected by H5N8 HPAI [4], but in 2021/2022 South Africa, Lesotho, Botswana and Namibia reported outbreaks of H5N1 HPAI. The WAHIS data available provides the basis for a broad overview at regional and global level, but precludes a detailed evaluation—for example, the Botswana outbreaks reported by Letsholo and coworkers [12] shows confirmed cases in the northwest region of the country where, by date of submission, this is not reflected in WAHIS reporting of the same outbreak (see https://wahis.woah.org/#/in-review/3924?fromPage=event-dashboard-url, accessed on 1 June 2023). The H5N1 events in South Africa occurred fairly early in the overall African outbreak context; however, this may be a result of a delay in reporting from other countries. 

South Africa has consistently been the most severely impacted by HPAI epidemics in the sub-region because it has the largest and most advanced poultry sector. However, fortunately it is also equipped with a network of accredited veterinary laboratories to make rapid molecular and serological diagnosis during field outbreaks of HPAI. For H/N subtyping and HPAI diagnostic purposes, the targeting of single IAV genes or gene portions is rapid and convenient, but the full viral genome sequence of ~15 kilobases is indispensable to accurate molecular epidemiological tracing, which was the purpose of this study. IAVs have the propensity to reassort during co-infections, which can provide direct data about the hosts or locations or origin, and secondly, even though an RNA virus IAV mutates relatively rapidly, the nucleotide substitution mutations still take several weeks or months to emerge, and by evaluating longer sequences, maximum genetic information can be obtained. Direct deep sequencing of high-titered clinical material precludes the need to isolate the virus for sequencing nowadays, but deep sequencing technology remains relatively expensive and beyond the reach of most developing countries as a diagnostic tool. Pivotal to the field of molecular epidemiology is access to a database of closely related viruses from other regions; therefore, the WOAH-FAO OFFLU scientific network encourages the sharing of IAV sequence data timeously. Viral genome data can be used to further our understanding of the disease, the hosts these viruses infect, and how and when the viruses translocate, ultimately supporting efforts to stabilize global food sources and protect biodiversity.

This study contributes >100 South African viral genome sequences to aid global H5Nx HPAI tracing and provides valuable insights into the regional spread of H5N1 HPAI in southern Africa in 2021–2022. According to the phylogenetic and genetic spatial analysis in this, as well as previous studies, West Africa receives HPAI H5N1 viruses circulating in Europe. The Senegalese strains to which the southern African viruses were most closely related were introduced by migratory birds from southern Europe in late 2020 [9] and moved southwards with intra-African migrant aquatic birds in the first month of 2021. The index cases of H5N1 HPAI in South Africa occurred in early April 2021, compared with the H5N8 HPAI index case in mid-May 2017 in Zimbabwe. The earliest outbreaks in South Africa in both 2017/2018 and 2021/2022 started in the same general north-central region of the country, the agriculturally intensive area along the Vaal River spanning regions of the North West, Gauteng and Mpumalanga provinces. The availability of water and cultivated crops (maize especially) in this region probably attracts migrant aquatic birds from West, Central and East Africa at this time of the year. Both the 2017/2018 and 2021/2022 H5Nx HPAI epidemics were characterized by an initial high viral diversity, suggesting a sudden influx of variants. Our analysis shows that seven distinct H5N1 HPAI sub-genotypes (SA1 to SA5, SA7 and SA8) were associated with outbreaks in wild birds and poultry in the first six weeks of the outbreaks in South Africa, but virus diversity decreased over the course of the epidemic, and by the end of 2022, only two sub-genotypes (SA13, SA15) were still circulating, according to our dataset. In 2021, H5Nx HPAI outbreaks in the Western Cape province started earlier than those in 2017, and unlike the 2017/2018 H5N8 HPAI epizootic where the outbreaks were associated with a single sub-genotype, a much higher virus diversity was present in the Western Cape H5N1 HPAI viruses of 2021/2022 (sub-genotypes SA3, SA4, SA5, SA6, SA8, SA9, SA10, SA10, SA13 and SA14). 

This phylogenetic study determined unequivocally that South African poultry was not the source of Lesotho’s poultry outbreaks, but that the latter were most likely point introductions from the wild bird reservoir, and that the causative strains (SA2) were likely introduced to the region concurrently with South Africa’s early sub-genotypes. Similarly, South Africa was not the source of Botswana’s outbreaks in wild birds and poultry in June 2021. Instead, a unique sub-genotype (SA15) sharing a progenitor with the Botswanan strains was introduced into the Free State province in the latter half of 2022 where it caused the outbreak in commercial ostriches in November. Speculatively, the reservoir for both South Africa’s SA15 and Botswana’s viruses is in central or East Africa, on the Black Sea–Mediterranean migratory flyway, which would account for the appearance of SA15 in the spring, when previous new introductions to South Africa occurred in the late summer/autumn ([4,28]; present results). Most of the outbreaks in South African commercial poultry, according to the phylogenetic analysis, were point introductions from wild birds. From the molecular evidence, we conservatively estimate that 7 out of 41 cases in commercial chickens were possible farm-to-farm spread, and therefore that 83% of cases in commercial poultry 2021/2022 were due to wild bird introductions or movement from the contaminated environment into the operations. This figure is similar to estimates by the United States Department of Agriculture [29] that less than 15% of the clade 2.3.4.4 H5N1 HPAI outbreaks in backyard and commercial farms in the USA in 2022 were caused by farm-to-farm spread. 

Another striking similarity between the 2017/2018 and 2021/2022 epizootics was the emergence of a coastal seabird-restricted sub-lineage (SA13) in the Western Cape province that also spread to Namibia. However, Cape cormorants died in the highest numbers (an estimated 24,000 at least) in 2021, whereas swift terns accounted for most deaths in 2018 [6]. The ages of affected Cape cormorants were not recorded but the numbers equate to 30% of the breeding population (A. Makhado, pers. comm.), so the virus could have had a significant effect on this endangered population. However, the same species bred without mishap in 2022 with the HPAI virus still circulating, so it is hoped that sufficient population immunity has developed. African penguins were affected in both the H5N8 and H5N1 HPAI epizootics, but in higher numbers in 2021/2022 than in 2018. With the remaining global population at less than 15,000 breeding pairs (Department of Forestry, Fisheries and the Environment unpublished census data, 2021; J. Kemper, pers. comm), the loss of more than 300 African penguins to H5N1 HPAI in 2021 and 2022 is concerning. There were deaths on Dyer Island in both 2021 and 2022, but far fewer in 2022, possibly due to the lack of virus in the Cape cormorant colony but hopefully also due to some accumulated herd immunity. Mass mortality events caused by HPAI in threatened or endangered wild bird species and breeding colonies of coastal seabirds have occurred in Europe, the Middle East and the Americas at an unprecedented scale, with a negative impact on conservation efforts and biodiversity [3,30]. Increasing numbers of infections are also being reported in mammals such as mink, foxes, skunks, raccoons, bears, sea lions, seals, domestic pigs and wild boars in Europe and the Americas [31], but no confirmed cases of H5N1 HPAI in marine mammals or other mammalian species have been made in southern Africa yet. H5N1 HPAI outbreaks continued in South Africa in 2023, and sequencing efforts are ongoing.

## Figures and Tables

**Figure 1 viruses-15-01383-f001:**
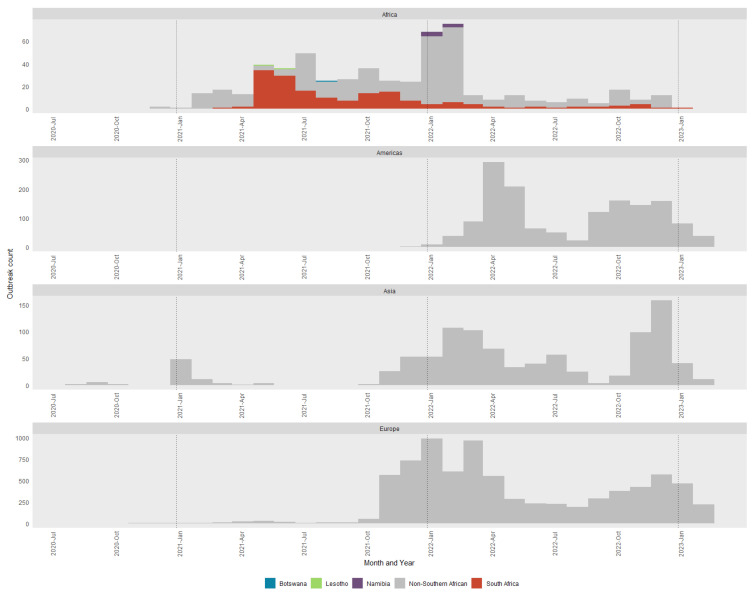
Monthly epidemic curve of all H5N1 outbreaks reported between July 2020 and February 2023 to the WOAH. Outbreaks are classified by the global region in which they occurred and colored by distinct countries if in southern Africa. The dotted lines represent January of each year. Note the scale on the *y*-axis differs by region.

**Figure 2 viruses-15-01383-f002:**
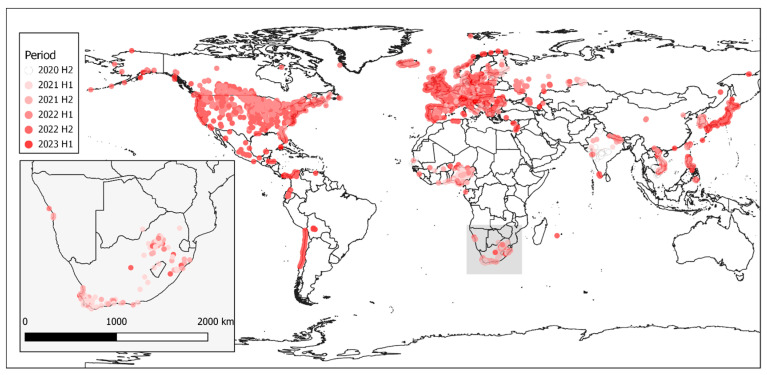
Map of all H5N1 outbreaks reported between July 2020 (2020 H2) and February 2023 (2023 H1) to the WOAH. Outbreak color reflects the half-year period from 2020 H1 through 2023 H1 [6 periods].

**Figure 3 viruses-15-01383-f003:**
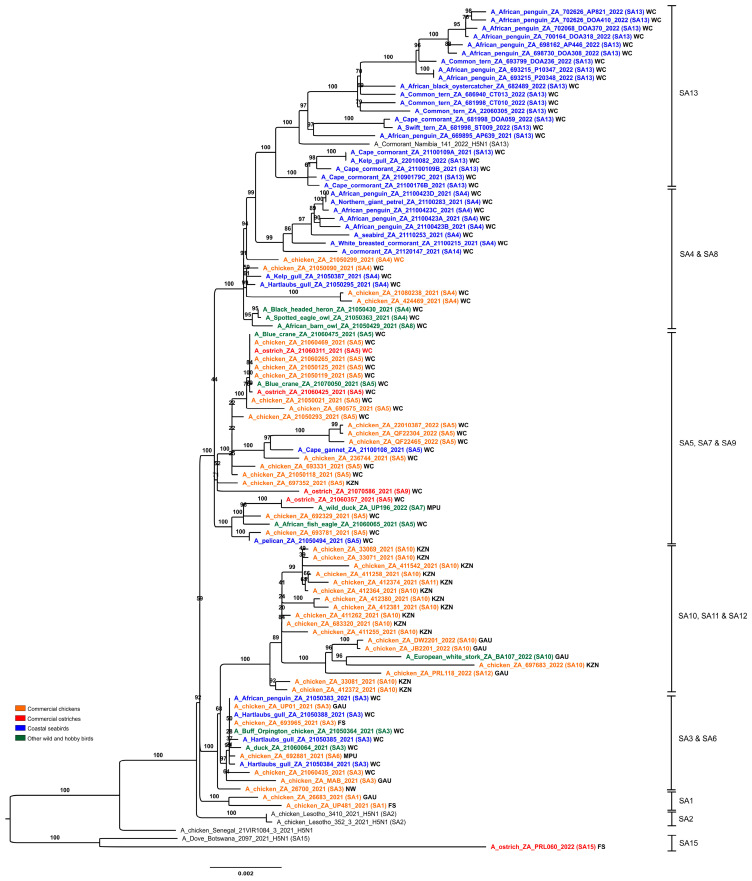
Maximum likelihood phylogenetic tree of the concatenated whole genomes of clade 2.3.4.4B H5N1 HPAI viruses from outbreaks in southern Africa in 2021–2022. Sequences generated in this study are in boldface, with sub-genotypes in brackets. Province abbreviations: WC—Western Cape; GAU—Gauteng; KZN—KwaZulu-Natal; NW—North West; MPU—Mpumalanga; FS—Free State.

**Figure 4 viruses-15-01383-f004:**
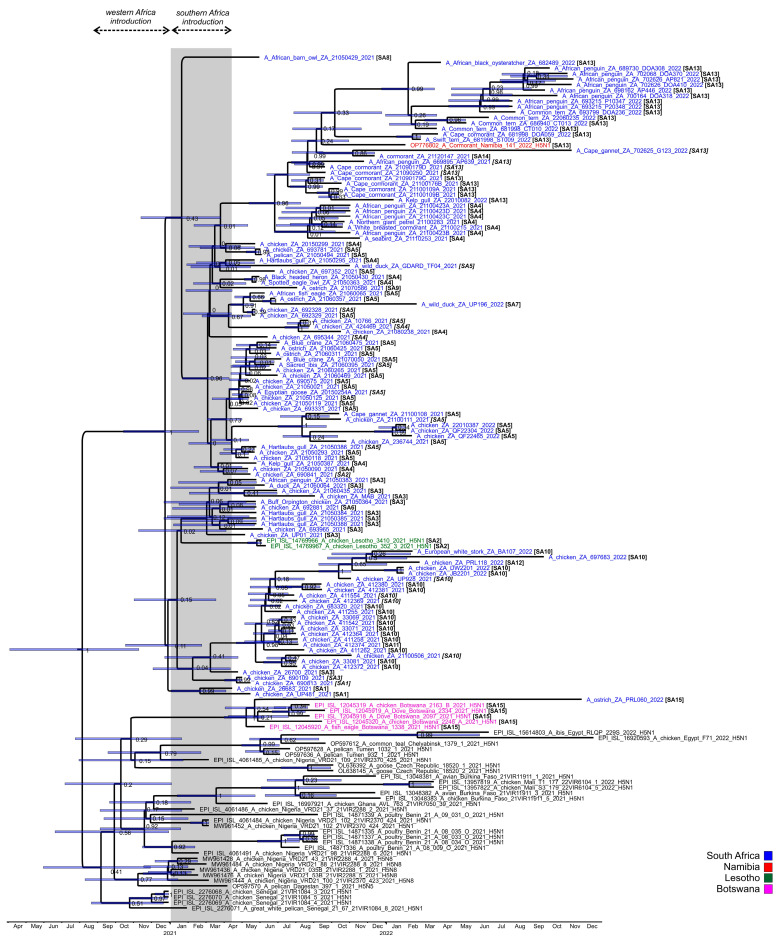
Maximum clade credibility tree of the H5 HA gene of clade 2.3.4.4B H5N1 HPAI viruses from southern Africa in 2021–2022 and the closest relatives. The nodes values represent the posterior probability and the bars represent the 95% highest posterior probability range. Square brackets denote the regional sub-genotypes assigned in this study, where these are italicized, the sub-genotype was inferred from a partial genome. The grey bar highlights the period between early January and April 2021 when the progenitors to the H5N1 HPAI viruses causing the southern African outbreaks emerged and spread southwards.

**Figure 5 viruses-15-01383-f005:**
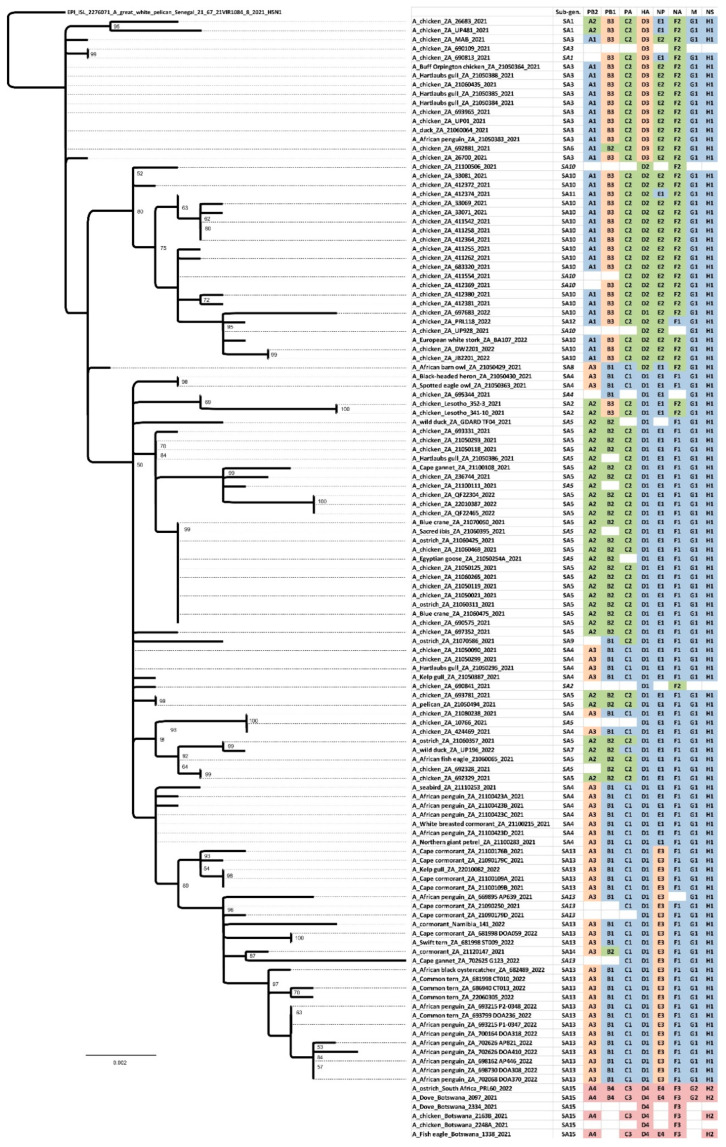
Designation of sub-genotypes SA1 to SA15 for clade 2.3.4.4B H5N1 HPAI viruses in southern Africa in 2021–2022. The maximum likelihood phylogenetic tree for the HA gene (SA1 to SA14 only) is rooted in the Senegalese virus; only bootstrap values ≥50% are shown. The sub-clades A1–4, B1–4, C1–3, D1–4, E1–4, G1 & 2 and H1 & 2 were inferred from the maximum phylogenetic trees in Appendix A. Italics represent sub-genotypes tentatively assigned from partial genome sequences.

**Figure 6 viruses-15-01383-f006:**
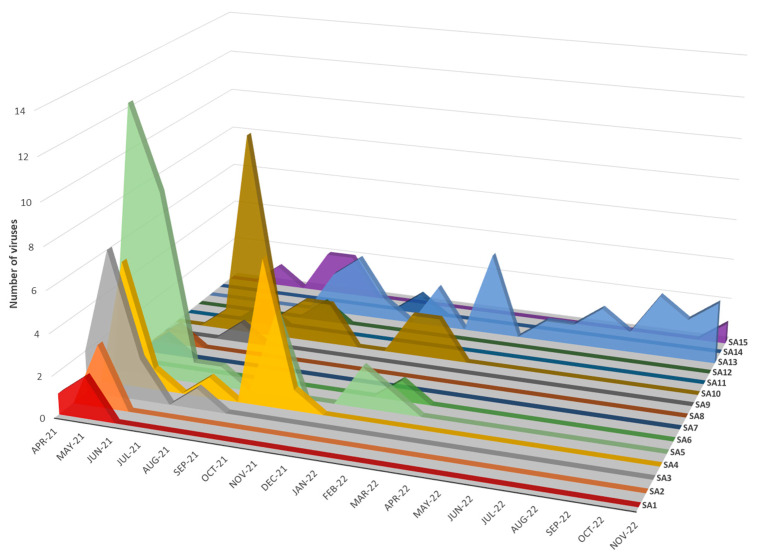
Temporal distribution of sub-genotypes SA1–SA15 detected in southern African clade 2.3.4.4B H5N1 HPAI outbreaks in 2021–2022.

**Figure 7 viruses-15-01383-f007:**
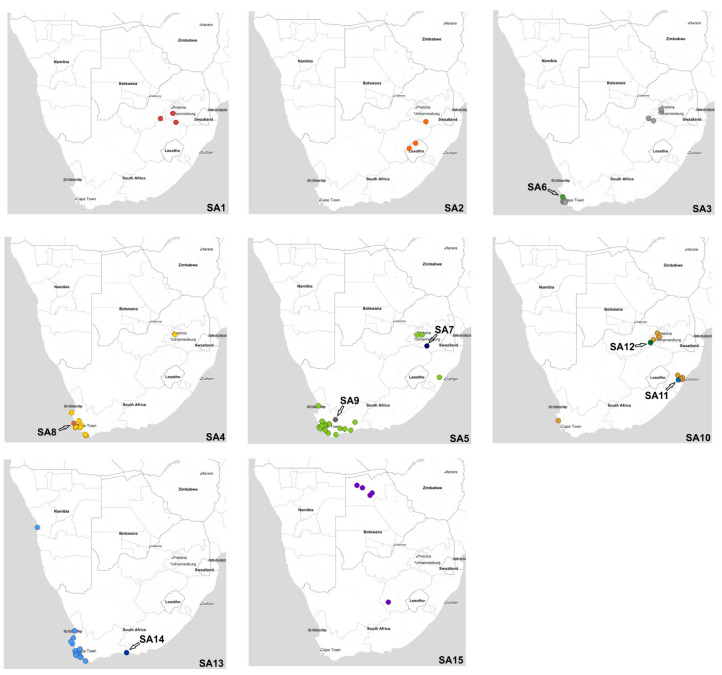
Geographic locations of clade 2.3.4.4B H5N1 HPAI sub-genotypes in southern Africa in 2021–2022. Sub-genotypes represented by a single strain are grouped with the closely related sub-genotype (SA3 and SA6), (SA4 and SA8), (SA5, SA7 and SA9), (SA10, SA11 and SA12) and (SA13 and SA14).

## Data Availability

All the data needed to generate the conclusions made in the article itself are present and/or in the Appendix A.

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
