# Peer review of "The Molecular Epidemiology of Clade 2.3.4.4B H5N1 High Pathogenicity Avian Influenza in Southern Africa, 2021–2022"

_viruses, 2023, doi:10.3390/v15061383_

Round 1
Reviewer 1 Report
I congratulate the authors on a well presented and comprehensive study of H5N1 HPAIVs that circulated in South Africa in 2021-2022. The data generated and the analyses undertaken fill in several knowledge gaps on the molecular epidemiology of this virus in the southern African region and will be of interest to many working in this field. I recommend publication following some minor revisions as listed below.
Although a very interesting read, I feel that many readers will find it too long and too detailed. I suggest that the authors try an summarise their findings as best they can.
Often with these studies and manuscript format, the quality and resolution of figures is poor. This is also the case here. Therefore, I suggest that all of the figures are also provided as pdf files (zoomable) in the supplementary materials.
Minor comments
(1) Page 3 line 115: please define "granular" in this context.
(2) Page 4 line 147. Please provide more details on the qRT-PCR protocol used.
(3) Page 4 lines 149 to 152. Please rewrite to improve clarity.
(4) Page 4 line 174. Please provide GISAID reference numbers.
(5) Figure 4. Is it possible to add the months to the x-axis?
(6) Figure 6. Please check to see if the text (Page 12 line 321 to 334) reflects what is shown in the figure.
(7) Page 16 line 492 to 495 Please rewrite to improve clarity.
(8) Page 17 line 521 should read "Sub-genotypes SA10, SA11 and SA12"
(9) Page 17 line 557 should read "cannot"
(10) Page 19 line 623 should read "Figure 3"
Author Response
I congratulate the authors on a well presented and comprehensive study of H5N1 HPAIVs that circulated in South Africa in 2021-2022. The data generated and the analyses undertaken fill in several knowledge gaps on the molecular epidemiology of this virus in the southern African region and will be of interest to many working in this field. I recommend publication following some minor revisions as listed below.
Although a very interesting read, I feel that many readers will find it too long and too detailed. I suggest that the authors try an summarise their findings as best they can.
Response: Thank you for the comment, yes, it is very detailed, but a lot of data was analyzed and there are important details in the text that are not reflected in the figures, for example the description of clinical signs in the ostriches. In our view, no information that isn’t important was included, but to reduce the length where possible, lines 641-648 on page 19 were revised.
Some other minor corrections to bird numbers and species were also made on pages 14-15 and 18-24 as indicated in the markup.
Often with these studies and manuscript format, the quality and resolution of figures is poor. This is also the case here. Therefore, I suggest that all of the figures are also provided as pdf files (zoomable) in the supplementary materials.
Response: We agree, PDF versions will be provided as suggested.
Minor comments
(1) Page 3 line 115: please define "granular" in this context.
Response: “Granular” refers to the fact that the population affected data was retrieved in the second run, it’s a lot more intensive to retrieve these data but it helped to look at differences between say the ostrich population affected between the two epizootics. The epidemiologist did not do this level of retrieval for the more general overview analysis. To avoid confusion, “granular” was changed to “detailed”.
(2) Page 4 line 147. Please provide more details on the qRT-PCR protocol used.
Response: Line 147 was amended to reflect that the qRT-PCR protocols are those recommended by the European Union Reference Laboratory, and the URL was provided (https://www.izsvenezie.com/reference-laboratories/avian-influenza-newcastle-disease/diagnostic-protocols).
(3) Page 4 lines 149 to 152. Please rewrite to improve clarity.
Response: Lines 149-150 were replaced with “…but samples from some backyard poultry and wild bird outbreaks were unavailable for analysis” to improve the clarity.
(4) Page 4 line 174. Please provide GISAID reference numbers.
Response: GISAID reference numbers are provided as requested (EPI_ISL_14542488 to EPI_ISL_1707211).
(5) Figure 4. Is it possible to add the months to the x-axis?
Response: Yes, months have now been added to the x-axis of Figure 4.
(6) Figure 6. Please check to see if the text (Page 12 line 321 to 334) reflects what is shown in the figure.
Response: Re-checked and we confirm that the text describes what is depicted in Figure 6.
(7) Page 16 line 492 to 495 Please rewrite to improve clarity.
Response: The sentence was rephrased to improve the clarity: “Viruses from outbreaks in other provinces were also genetically distinct, and therefore point introductions into those operations, examples include … on 27 August).
(8) Page 17 line 521 should read "Sub-genotypes SA10, SA11 and SA12"
Response: Corrected as suggested
(9) Page 17 line 557 should read "cannot"
Response: Corrected as suggested
(10) Page 19 line 623 should read "Figure 3"
Response: Corrected as suggested
Reviewer 2 Report
This article is devoted to the study of the H5N1 virus in southern Africa. The article summarizes a large amount of material on outbreaks of this virus during 2021-2022 years. Given the importance of this infection for the economy of this region and the world as a whole, this article is of great interest to veterinary specialists, virologists and zoologists. It is written in clear and understandable language, well illustrated and can be published in the journal without changes.
Author Response
Thank you very much, no revisions to add.